# Biomonitoring of Alterations in Fish That Inhabit Anthropic Aquatic Environments in a Basin from Semi-Arid Regions

**DOI:** 10.3390/toxics11010073

**Published:** 2023-01-12

**Authors:** Juan Manuel Pérez-Iglesias, Nadia Carla Bach, Patricia Laura Colombetti, Pablo Acuña, Jorge Esteban Colman-Lerner, Silvia Patricia González, Julie Celine Brodeur, Cesar Américo Almeida

**Affiliations:** 1Laboratorio de Química Analítica Ambiental (LAQUAA), Instituto de Química de San Luis (INQUISAL-CONICET), FQByF, UNSL, Ejército de Los Andes 950, San Luis D5700, Argentina; 2Departamento de Ciencias Ambientales y Producción, Universidad Nacional de Los Comechingones, Héroes de Malvinas S/N, Merlo, San Luis D5881, Argentina; 3Área de Biología, Facultad de Química, Bioquímica y Farmacia (FQByF), Universidad Nacional de San Luis (UNSL), Ejército de Los Andes 950, San Luis D5700, Argentina; 4Centro de Investigación y Desarrollo en Ciencias Aplicadas “Dr. Jorge J. Ronco” (CINDECA), La Plata B1900, Argentina; Consejo Nacional de Investigaciones Científicas y Técnicas, (CONICET), La Plata B1900, Argentina; 5Instituto de Recursos Biológicos, Centro de Investigaciones de Recursos Naturales (CIRN), Hurlingham B1686, Argentina; 6Instituto Nacional de Tecnología Agropecuaria (INTA), Hurlingham B1686, Argentina

**Keywords:** bioindicator fish, *Jenynsia multidentata*, habitat fragmentation, physiological stress, water quality

## Abstract

Industrial, agricultural, and urban areas can be sources of pollution and a cause of habitat fragmentation. The Conlara River located in the northeast of San Luis Province suffers different environmental pressures along its course from urban to agro-industrial areas. The present study aims to assess the water quality of the Conlara basin by evaluating how metals and pesticide contamination as well as physicochemical parameters relate to physiological stress in *Jenynsia multidentata*. Samplings were carried out in four sites characterized by a growing gradient of anthropic impact from the springs to the final sections of the river, starting with tourism passing through urban areas and ending with large agricultural areas (from S1 to S4) during both the dry and wet seasons. A total of 27 parameters were determined (11 physicochemical, 9 heavy metals, and 7 pesticides) in surface waters. Biomarkers (CAT, TBARS, ChE, and MN) showed significant physiological and cytological alterations in *J. multidentata* depending on the hydrology season. The combination of physicochemical parameters, metals, and pesticide levels allowed typification and differentiation of the sites. Some metal (Cr, Mn, Pb, and Zn) and pesticide (α-BHC, chlorpyrifos, permethrin and cypermethrin, and endosulfan α) levels recorded exceeded the recommended Argentinian legislation values. A principal component analysis (PCA) allowed detection of differences between both seasons and across sites. Furthermore, the differences in distances showed by PCA between the sites were due to differences in the presence of physicochemical parameters, metals, and pesticides correlated with several biomarkers’ responses depending on type of environmental stressor. Water quality evaluation along the Conlara River shows deterioration and different types of environmental stressors, identifying zones, and specific sources of pollution. Furthermore, the biomarkers suggest that the native species could be sensitive to anthropogenic environmental pressures.

## 1. Introduction

Environmental integrity is the capacity of the environment to maintain a balanced and integrated adaptive community of living organisms, given species composition, diversity, and functional organization [1,2]. One of the most important factors defining environmental integrity in an aquatic ecosystem is water quality, which can be evaluated through the determination of various physical, chemical, and biological parameters [3,4]. The quality of surface waters is determined, to a certain degree, by the nature and extent of anthropogenic activities in its catchments [5,6]. The rapid development in terms of industrial and agricultural activities has raised severe concerns about the quality of aquatic ecosystems, due to releases of stressors from natural components (e.g., weathering of parent rocks, volcanic eruption) as well as from various human activities (e.g., industries, intensive agriculture practices, sewage sludge, municipal waste, landfill, smelting, and mining) [7]. Meanwhile, urbanization is associated with an increase in impervious surfaces (e.g., roads, parking lots, etc.), which impacts water quality by altering the frequency and magnitude of overland runoff, increasing erosion and transporting pollutants to stream channels [6]. In this sense, urban, industrial, and agricultural activities can negatively impact water resources and, in consequence, affect aquatic life [8,9]. In other words, land use changes due to anthropogenic activities alter the spatial structure (composition and configuration) of original landscapes and affect aquatic and terrestrial biodiversity negatively through habitat loss and fragmentation [10].

Recently, studies suggest that pressures on aquatic ecosystems have increased due to the widespread presence of environmental stressors affecting the normal functioning of living organisms [2,6,9,11,12,13]. The longitudinal connectivity along the stream or river (from the lower to the upper parts of the basin) can be compromised or interrupted by anthropic activity, creating a fragmentation of the aquatic habitat, similar to what can be found in a dammed site, but produced by a pollution barrier [2,14]. In the same line, environmental pollution is considered an important factor that fragments habitats because it limits the flow between species, for example, making it possible for those that are more tolerant to occur only in regions of the river that are more polluted or anthropized [14].

It is known that continuous monitoring is one of the most reliable practices for obtaining information regarding the quality of natural water resources. In this regard, water quality assessment was early dominated by standardized physicochemical monitoring [9,12]. However, this approach does not allow integration of all the pollutants affecting the aquatic environment. For this reason, ecotoxicological biomonitoring practices have increased in recent years and have been added to monitoring action plans [6,9,12,13]. Indeed, the combination of biomonitoring and ecotoxicological evaluations results in an interesting tool to thoroughly assess water quality. In this sense, biomonitoring biomarkers complement and enhance the reliability of chemical analysis, offering a more comprehensive analysis of the potential effects of stressors in organisms [9,15]. Variations in these biomarkers can reflect effects on water bodies and can be used to analyze changes in water quality at different sites and/or in different periods [8,9]. Bioindicators can determine the overall effects of environmental stressors within the complexity of natural systems. Biochemical and cellular biomarkers, such as reactive oxygen species (ROS) leading to oxidation of DNA and proteins (enzymes) and physiological stress responses, may determine the potential of environmental stressors, directly or by means of toxicokinetic and toxicodynamic studies [16,17].

Among freshwater environments, small streams and rivers are most adversely affected. This is probably due to their position within the landscape, where they usually serve as receivers of waste, sediments, and pollutants in runoff and infiltration processes [18]. Particularly, in semiarid regions, rainfall precipitation is highly variable but is a major factor controlling the hydrological cycle of a region and, thereby, also ecological and geomorphologic processes [19,20]. These characteristics determine the presence of a hydrological cycle with two markedly different seasons. The “high water” or wet season that occurs during spring–summer is characterized by intense rainfall (between 70 and 110 mm), and a period of “low water” or dry season occurs during autumn–winter, characterized by low rainfall (between 8 and 40 mm). In addition, the water reservoirs in arid and semi-arid regions are important to maintaining the local flora and fauna, as well as being a major source of drinking water for local populations [7,21].

In this context, the present study aimed to assess the water quality of the Conlara basin (San Luis, Argentina) by evaluating how metal and pesticide contamination as well as physicochemical parameters relate to physiological stress (individual, biochemical, and cytogenetic endpoints) in the native fish *Jenynsia multidentata*.

## 2. Materials and Methods

### 2.1. Study Site

Located in Argentina, the Conlara basin covers part of the provinces of San Luis and Córdoba, with an area of around 8800 km^2^. This basin is a tectonic depression trending North–South, flanked by crystalline blocks that constitute Sierra de Los Comechingones to the east and San Luis to the west, filled with erosional material from high regions and to a lesser extent by loess. It is currently an urban–tourist region of great importance and it belongs to the most intense and important agricultural region of San Luis Province.

Four sampling points were selected on the Conlara River, (Figure 1). Site 1 (S1) is located upstream near Paso Grande town, a small city with 427 inhabitants with low livestock and tourist activity during summer (S1: 32.8794191 S, 65.635656 W). Site 2 (S2) is at the confluence with the river of Renca town, a small city with 178 inhabitants with low livestock and tourist activity during summer (S2: 32.7759708 S, 65.3642264 W), although upstream of the river that crosses the Renca town (S2) there is an artificial dam where tourism and fishing activities take place. Site 3 (S3) is near a river located in a high-impact urban and tourist area, corresponding to the town of Concarán (S3: 32.5604637 S, 65.2551778 W) with a population of 5100 inhabitants, and receives water with contaminants from local agricultural activity and industrial activity from five upstream towns. Finally, site 4 (S4) is located in an area of high agricultural activity, near the town of Santa Rosa de Conlara (S4: 32.3353004 S, 65.2125166 W) with a population of 5500 inhabitants, and receives impacts from industrial, agricultural, tourism, and urban activities from upstream towns.

### 2.2. Chemical Determinations: Physicochemical, Metals, and Pesticides

Monthly water samples were collected over one full year, in the period ranging between March 2020 and March 2022. Sampling, preservation, and transportation of the water samples, and physicochemical determinations, were performed according to Standard Methods for the Examination of Water and Wastewater [22]. The reagents used were of analytical quality. Distilled water with a conductivity lower than 2 µS·cm^−1^ was used to prepare solutions and to dilute samples when necessary. All parameters were determined in the laboratory. The specific methods used for the different physicochemical parameters were as follows: pH S.M. 4500-B; conductivity (µS·cm^−1^) S.M. 8 2510-B; nitrates (mg L^−1^) S.M. 4500 NO_3_^−_^E; nitrites (mg L^−1^); S.M. 4500-NO_2_^−^-B; ammonium (mg L^−1^) S.M. 4500-F-NH^4+^; phosphates (mg L^−1^) S.M. 4500—PO_4_^3−^ E; chemical oxygen demand—COD (mg L^−1^) S.M. 5220-B; biological oxygen demand—BOD (mg L^−1^) S.M. 5210-B; organic matter (mg L^−1^ O_2_) S.M. 5220-D; dissolved oxygen S.M. 4500-0-C (mg L^−1^ O_2_); total hardness (mg L^−1^) S.M. 2340 C. Metal determination of arsenic (As), barium (Ba), cadmium (Cd), copper (Cu), chromium (Cr), iron (Fe), manganese (Mn), lead (Pb), and zinc (Zn) was performed by Inductive Coupled Plasma Mass Spectrometry (ICP-MS). For pesticide determination, all the solvents used were from J.T. Baker (Phillipsburg, NJ, USA) and the standards were Chem-Lab (Zedelgem, Belgium) (lot 20.3322302.25). All standard solutions of organochlorine and organophosphate pesticides were prepared with the respective EPA 507 and 508 standards (Method 3500) [23]. Water samples pre-treatment (250 mL) consisted of liquid–liquid extraction with dichloromethane (three contacts), rotary evaporation, and drying with N_2_ flow and resuspension in 1 mL of n-Hexane (Method 3500) [23]. Water samples were processed and determined by gas chromatography (CG Agilent 6890N) with a μECD detector, using an HP-5 capillary column (15 m × 0.53 mm × 1.50 μm), H_2_ as carrier gas, and N_2_ as make-up, with a temperature program of 150 °C for 1 min and ramp at 4 °C min^−1^ up to 260 °C, maintaining it for 4 min.

### 2.3. Test Organisms and Specimen Recollection

In this work, the one-sided livebearer *Jenynsia multidentata* (Anablepidae, Cyprinodontiformes) was used for biomonitoring. A total of ten individuals (*n* = 10) were collected per site per sampling season. This fish is a native viviparous species that presents external sexual dimorphism. It is widely distributed and abundant in the Neotropical region of South America in both polluted and non-polluted areas [10,24,25]. *J. multidentata* has been proposed for use as a bioindicator organism in environmental studies covering various regions of South America [11,12,24,25,26]. Added to this, several laboratory investigations have used *J. multidentata* as a bioindicator of toxic substances [11,27,28,29,30,31,32,33]. Finally, we selected the species not only because it is widely used in biomonitoring in the region, but also because it is abundantly present at all sampling sites.

All fish were collected under a permit emitted by the government of San Luis Province, Argentina (Resolution 49-PMA2019). All animal manipulations were performed according to the “Reference Ethical Framework for Biomedical Research: Ethical Principles for Research with Laboratory, Farm, and Wild Animals” [34]. Fish were collected using traps and transported to a sample processing location in a bucket containing aerated water from the sample site. Every fish was anesthetized, photographed, weighed, and measured (total length), followed by cutting the neural cord behind the brain according to the previously mentioned protocols [25,34]. A blood sample was taken from each animal by opening the operculum and the entire animal was placed in 15 mL conical flasks containing PBS at 4 °C to be finally stored at the same temperature until enzymatic analysis in the laboratory [25].

### 2.4. Biological Endpoints

#### 2.4.1. Individual Endpoints

Fish body condition (K) was assessed using the method described by Schulte-Hostedde et al. [35]. This method consists of examining the residuals from a regression of body mass against snout-vent length where the regression line obtained establishes the average body weight for a given length. Then, an individual with positive residuals is considered to be in a good condition, whereas an individual with a negative residual is regarded as having low energy [35]. Individuals were weighed to obtain the body condition using a precision scale of 0.001 g and snout-vent length was determined by photograph analysis with the ImageJ program, freely available online (https://imagej.nih.gov/ij/index.html, accessed on 1 February 2022). The calculation of the K value is obtained by the quotient between the weight and the length raised to the cube or to the third power.

#### 2.4.2. Biochemical Endpoints

All procedures were performed as previously described by Brodeur et al. [25]. Whole fish were homogenized in an ice-cold 50 mM tris buffer (1 mM EDTA acid, 0.25 M of sucrose, pH 7.4) with a Teflon-glass Potter–Elvehjem homogenizer. Then, the homogenates were centrifuged at 4 °C (10,000× *g*, 10 min) to collect the supernatant while nuclei and cell debris were discarded. All biochemical enzyme reactions and protein determinations were measured using a spectrophotometer (Rayleigh—Model UV2601 UV/VIS Double-Beam Spectrophotometer, Beijing, China).

##### Protein Determination

A portion of the supernatant was used to determine protein concentrations using the Bradford method [36]. Bovine serum albumin (BSA) was used as a standard.

##### Determination of Catalase Activity

Catalase (CAT) activity was determined by measuring the decomposition of hydrogen peroxide at 240 nm (37 °C, 2 min), using a molar extinction coefficient of 43.6 M^−1^·cm^−1^ [37]. The reaction mixture consisted of 20 μL of a pure sample, 40 μL of H_2_O_2_ (10%, *v*/*v*), and 1900 μL of PBS (pH 7, 100 mM).

##### Lipid Peroxidation by Thiobarbituric Acid Reaction

The lipid peroxidation was determined by the reaction of thiobarbituric acid-reactive substances (TBARS) according to the method of Buege and Aust [38], with minor modifications for fish. The lipid peroxidation in whole fish was determined by measuring the formation of the color produced during the TBARS reaction. To this end, fish homogenate (20 μL) and 380 µL of the reaction mixture (trichloroacetic/thiobarbituric acid) were incubated at 90.0 ± 0.5 °C for 15 min; then, the colored product was cooled and centrifuged at 7500× *g* for 8 min. Finally, the absorbance was measured at 530 nm. Lipid peroxidation or TBARS levels were expressed as mmol MDA mg^−1^ protein [38].

##### Determination of Cholinesterase Activity

Cholinesterase activity (ChE) was determined by the method of Ellman et al. [39]. The reaction mixture consisted of 150 µL of PBS (100 mM, pH 8), 50 µL of acetylthiocholine iodide (1 mM), 150 µL of 5,5′-dithiobis-(2-nitrobenzoic acid) (0.5 mM), and 10 µL of a pure sample. The change in absorbance was recorded at 412 nm (37 °C, 1 min). The enzymatic activity was calculated using a molar extinction coefficient of 14.150 M^−1^ cm^−1^.

#### 2.4.3. Cytogenetic Endpoints

##### Micronucleus (MN) Induction

MN assay was conducted following the original protocol [40] with minor modifications for Neotropical fish [41]. Slides of blood smears obtained in the field by extracting blood from ten fish by site were stained for 12 min with 5% of Giemsa solution for each treated group. MN frequency was calculated in peripheral mature erythrocytes after acute pulse exposure in both scenarios. MN were analyzed from 1000 mature erythrocytes on each blood fish sample (×1000 magnification). MN frequencies are expressed as a total number of alterations per 1000 cells and the examination criteria for MN acceptance were determined following previous reports [42].

### 2.5. Statistical Analysis

A one-way analysis of variance (ANOVA) followed by a Tukey test was performed to evaluate the differences among sites of the following parameters (response variables): body condition (K), enzymes activities (CAT, TBARS, and ChE), frequencies of MN. Normality and equal variance were corroborated by Chi Square and a Bartlett test, respectively [43]. In some cases, data had to be logarithmically transformed to meet assumptions of normality and equal variance. A non-parametric Kruskal–Wallis test was performed in cases where the latter assumptions could not be met [43].

Principal component analysis (PCA) was performed considering each site as a grouping variable to improve the interpretation of the results and to obtain a holistic vision and data information of the biomarker responses [44,45]. In addition, the relationship between the biomarkers and sites were evaluated with a correlation matrix (Pearson product-moment correlation coefficient) by using simple linear regression [44,45]. Tests of significance of the regression and correlation coefficients were performed according to Zar [43]. The level of significance was α = 0.05 for all tests, unless otherwise indicated. Analyses were performed using the R software 6 v. 2.11.1 (R Core Team 2010).

## 3. Results

### 3.1. Chemical Determinations

The analytes detected in each site are represented in Table 1. The determinations allowed collection of data on the presence of 26 analytes, both in the dry season as well as wet season.

### 3.2. Biological Endpoints

#### 3.2.1. Individual Endpoints

Evaluations of body condition in *J. multidentata* did not reveal differences among sites (*p* > 0.05) between hydrological seasons (Figure 2).

#### 3.2.2. Biochemical Endpoints

Results of biochemical activities from fish collected at S1 relative to those collected at the sites potentially more impacted (S2, S3, and S4) are presented in Figure 3. In particular, an increase in CAT activity was observed only in the wet season for S4 with respect to S1, S2, and S3 (*p <* 0.05, Figure 3B). On the other hand, TBARS showed a significant difference in S2, S3, and S4 during the dry season (*p <* 0.05, Figure 3C) compared to S1. Furthermore, in the wet season, sites S3 and S4 showed a significant difference only with respect to S1 (*p <* 0.05, Figure 3D). In addition, ChE was statistically significant, with an increase in S3 and S4 compared to S1 and S2 in the dry season (*p <* 0.05, Figure 3E). In the wet season, a significant increase was observed in ChE activity in S1 compared to S4 (*p <* 0.05, Figure 3F).

#### 3.2.3. Cytogenetic Endpoints

Cytological responses revealed a significant increase in MN in fish from S3 compared to those from S1, although this observation was only made during the wet season (*p <* 0.05) (Figure 4).

### 3.3. Integration of Environmental Stressors and Biological Measurements by Multivariate Analysis

The results of the grouping of the sites in function of the analyzed variables are shown in Figure 5. For reduction in the number of variables, chemical variables that had a high positive correlation according to the correlation matrix were grouped into the same variable called PC (physicochemical variables) or GM (metals). Furthermore, the biological responses (biomarkers) that showed significant differences between sites were selected (excluding the K index for both seasons, and CAT and MN in the dry season). The variables employed in the multivariate analysis allowed us to explain, in the dry season, 98.70% (PC1: 63.73%, PC2: 34.97%) of variability, while in the wet season the analysis explains 92.46% (PC1: 61.02%, PC2: 31.44%) of the variation. In general, for both hydrological seasons, the analysis showed that the sites throughout the basin are different from each other and this is due to the different contamination that occurs among sites (Figure 5).

In general, during the dry season, the four sites were different. In particular, S1 was characterized by the presence of GMD2 (second metal group correlated during dry season: cadmium, copper, barium, and lead) and normal values of biomarkers. S2 was a transition between S1 and S3, and was characterized by similarity with S3 with high concentrations of phosphates. On the other hand, S3 and S4 showed similarity and correlations in terms of greater values for the physicochemical variables of group 1 (PCD_G1), whereas S4 was different from the rest of the sites by the presence of the highest concentrations of endosulfan sulphate, cypermethrin, and GMD1 (first metal group correlated during dry season: arsenic and magnesium). Finally, S3 and S4 were different from S1 because of alterations in biomarkers of physiological response (Figure 5A).

During the wet season, large differences between sites were observed (Figure 5B). In S1, the difference with other sites was explained by the highest concentrations of chlorpyrifos, endosulfan alpha, GMW2 (second metal group correlated during wet season: chromium and zinc metals), and with biomarkers alterations. In addition, in S2 there was a similarity with S3 but there were also differences between sites due to the high concentrations of phosphate in S2, while the highest values of GM3 (third metal group correlated during wet season: Fe and Ba) led to biomarker response MN. Finally, in S4, results were very different from the rest of the sites, with separation due to the presence of high concentrations of PCW_G1 (first group of physicochemical variables correlated during wet season: pH, nitrate, nitrite, ammonia, phosphate, total hardness, organic matter, dissolved oxygen), COD, endosulfan sulfate, cypermethrin, and GMW1 (first metal group correlated during wet season: As and Pb metals) causing the response of CAT, ChE, and TBARS biomarkers.

## 4. Discussion

This work demonstrates the alteration of the aquatic habitats due to different types of environmental stressors, which depends on human activities, from upstream to downstream of the Conlara River in the Conlara basin. In this regard, recent studies have demonstrated that anthropogenic activities may cause border effects in a riparian forest or a pollution barrier with habitat fragmentation [2,10,14]. This situation, in the last instance, leads to decreases in the biodiversity of aquatic vertebrates at sites with environmental contamination [2,10,14]. Added to this, in recent years several studies around the world have shown that water reservoirs are contaminated with important environmental stressors due to anthropic activities affecting fish through their mode of action and/or bioaccumulation breaking trophic relationships [2,7,13,25,46]. In this context, it is urgent to develop early warning tools such as biomarkers that allow environmental damage to be detected before it becomes irreversible. Therefore, it is worth mentioning that, in agreement with previous studies, the present study demonstrates the usefulness of integrating physicochemical and biological biomonitoring to detect environmental alterations leading to pollution and eventually habitat fragmentation.

In effect, as previously mentioned, in semi-arid mountain environments where water is a resource on which all the biodiversity of the region depends, it is important to consider the hydrological dynamics not only of the bodies of water but also of the pollutants, because it has been shown that they produce different responses in bioindicator organisms [8,19,20]. In this sense, and in accordance with the latter authors, we here demonstrate that different environmental stressors for the biota should be considered in the wet or dry seasons. On the other hand, comparing with previous works in Neotropical regions, we can observe that our results coincide notably with those reported by Brodeur et al. [25] and Ballesteros et al. [11] for the same study species (*J. multidentata*). Particularly, the authors report alterations in antioxidant systems (CAT and TBARS) which are related to an increase in ROS outside the cell medium [9]. Moreover, our results showing ChE inhibition in the wet season, are related to the presence of organochlorine and organophosphate insecticides such as those reported previously for *J. multidentata* [11,25]. Furthermore, significant increases in ChE in the dry season are dependent on the presence of metals [11]. It is important to note that TBARS showed a high correlation with metals, indicating that it could be a useful biomarker for metal exposition. In this sense, aquatic invertebrates exhibit induction of CAT and TBARS as a defense against heavy metals and pesticides, to mediate biotransformation of xenobiotics and detoxification from hydrogen peroxide [47]. Added to this, the alterations of the cholinergic system (estimated in this study by the ChE activity) are signs of risk due to contamination [47]. In total accordance with these reports, we consider the possibility that the negative effects of environmental contamination may be mediated by a disruption of the oxidative balance, decreasing adaptability and contributing to reduced fitness of aquatic organisms [11,47]. Furthermore, these findings provide a more complete picture of the oxidative status of individuals threatened by habitat fragmentation [47]. However, as some authors mention, further studies should include the measurement of more antioxidant parameters as well as other aspects of oxidative stress (e.g., oxidative DNA damage). On the other hand, when we compare our results with other studies carried out on bioindicator species of fish used in the Neotropical region, such as *Cnesterodon decemmaculatus*, we observed similar results. For example, in studies where biomarkers are used at the individual, biochemical, and cellular levels for laboratory bioassays carried out with samples from rivers and effluents, alterations were observed only in cellular and biochemical biomarkers due to physicochemical parameters and metals [48,49,50,51,52]. Added to this, other laboratory studies with river water corroborate that water toxicity varies widely both spatially and temporally, suggesting intermittent toxic spillage, as confirmed by analyzing variations in the measured sample physiochemical parameters [53]. In particular, it is interesting that the investigations of recent years propose evaluation in fish of the same biomarkers used in our work for biomonitoring of surface water quality [52].

In turn, the multivariate analysis allowed us to verify differences between sites based on both chemical contamination and the biological endpoints that responded in each situation. A differentiation among sites was observed depending on the type of contamination observed as the basin advances towards more anthropized sites, in both hydrological seasons considered during a year. Similar results were reported by Ouyang et al. [8] and for mountain environments [11]. It is worth mentioning that previous studies and the present study signal the importance of using different biomarkers in biomonitoring since they allow the detection of differential effects of environmental contaminants. On the other hand, fragmentation mostly affects habitat quality as well as connections between habitat patches, and also limits the presence of sensible species [14]. In the present work, we observed that there is physiological stress in *J. multidentata,* depending on the degree and type of contamination, and verified by the response of different cytogenetic and/or biochemical biomarkers evaluated in the fish along the course of the river. At this point, our work shows that, in native fish, the cost to survive in polluted sites implies high physiological maintenance that results from cytogenetic damage to biochemical alterations. In the multi-stressor context, contaminants can affect the structure and function of biological systems, causing responses (biomarkers) at molecular, biochemical, histological, and behavioral levels before the community level is affected [11]. In this context, eventually, this can lead to the death of fish or elimination of sensitive species, disrupting biological communities throughout the basin. In sum, the findings of the present study agree with those of other authors [11,24,25,32,33], highlighting that *J. multidentata* is a bioindicator species not only in semi-arid regions but also in the Neotropical region. It is also demonstrating that this bioindicator species is affected in its fitness by different sources of contamination detected by early warning systems (biochemical and cytological biomarkers).

Finally, we emphasize that future studies in the region should focus on the potential changes in abundance or richness at the population and community levels in order to understand the impact of habitat fragmentation produced by environmental stressors in mountain aquatic ecosystems and from semi-arid regions. Other authors have highlighted that these environmental stressors, such as heavy metals, can bioaccumulate in local fish and put human populations at risk [7,22,54]. In our particular work, we found that the concentrations of metals such as Cr, Mn, Pb, and Zn, at all sites, are above the water quality guide levels for the protection of aquatic life according to Argentine legislation which is based on U.S.EPA recommendations [55], and in the case of Cr and Mn even above the water quality guide levels provided by Argentine legislation for human consumption [55]. On the other hand, when evaluating the situation of pesticides in all the sites, it is observed that the values of α-BHC, chlorpyrifos, permethrin and cypermethrin, and endosulfan α is above the water quality guide levels for the protection of aquatic life by Argentine legislation and Canadian Council of Ministers of the Environment legislation [55,56]. In this context, we consider that aquatic biota in this semiarid region is not only endangered by environmental stressors, but that risk assessments should be carried out on the potential effects of these environmental stressors, considering not only aquatic species but also bioaccumulation, the presence in compartments of sediments, and effects on the human population that uses these resources, using an integration of strategies between biomonitoring, biomarkers, and analytical determinations in ecosystem compartments for better decision making.

## Figures and Tables

**Figure 1 toxics-11-00073-f001:**
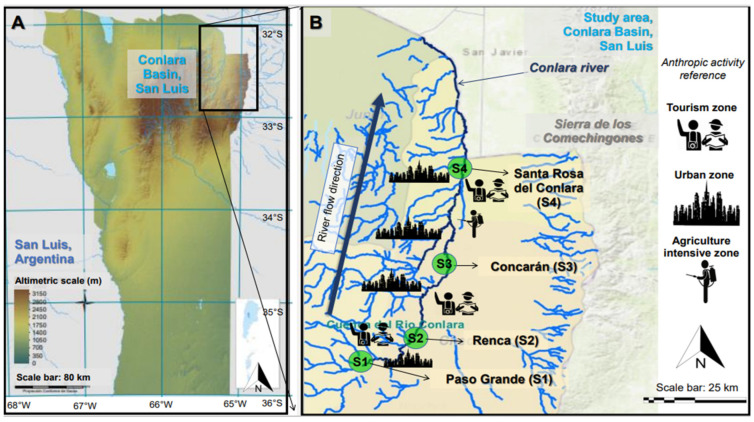
Map of San Luis Province, Argentina (**A**) showing in detail (**B**) the four study sites (S1–S4) at the Conlara River and the nearby anthropic activities within the Conlara basin from San Luis, Argentina. The studied Conlara River is represented with a dark blue color within the Conlara basin. The legend of the iconographies is on the right of (**B**).

**Figure 2 toxics-11-00073-f002:**
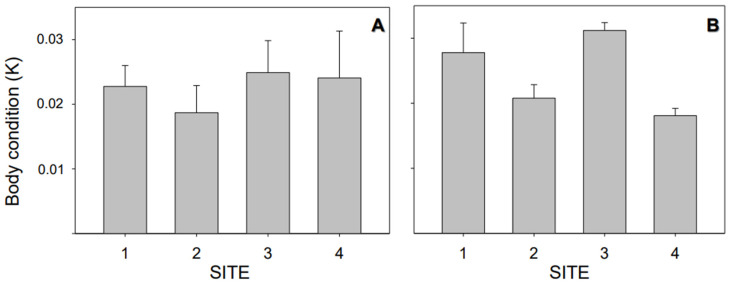
Individual endpoint measured in *Jenynsia multidentata* collected at the four study sites (S1 to S4), with body condition based on residual analysis in the dry (**A**) and wet (**B**) seasons. The whiskers represent the ± SEM.

**Figure 3 toxics-11-00073-f003:**
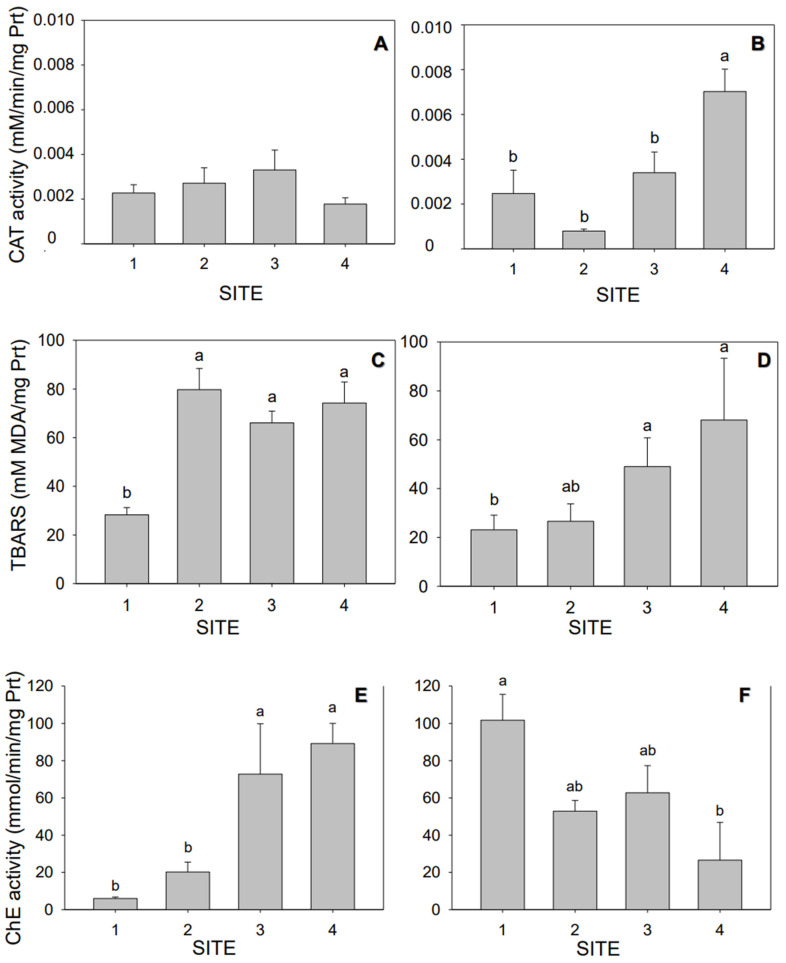
Biochemical endpoints measured in *Jenynsia multidentata* collected at the four study sites (S1 to S4), as catalase (CAT; (**A**,**B**)), lipid peroxidation (TBARS; (**C**,**D**)), and cholinesterase (ChE; (**E**,**F**)) in the dry (**A**,**C**,**E**) and wet (**B**,**D**,**F**) seasons. Different letters indicate statistically significant differences among sampling sites (*p* < 0.05). The whiskers represent the ± SEM.

**Figure 4 toxics-11-00073-f004:**
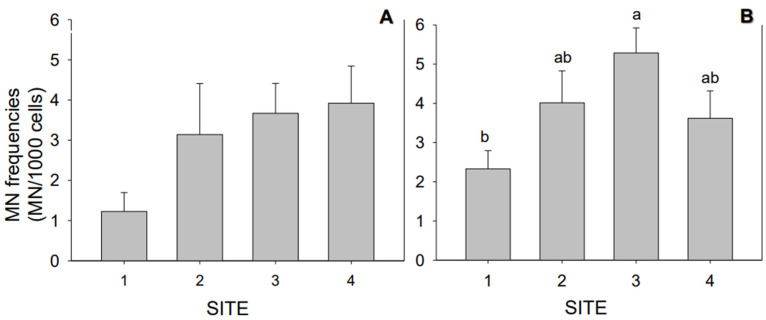
Cytological endpoint measured in *Jenynsia multidentata* collected at the four study sites (S1 to S4), as micronucleus (MN) induction in the dry (**A**) and wet (**B**) seasons. Different letters indicate statistical differences among sampling sites (*p* < 0.05). The whiskers represent the ± SEM.

**Figure 5 toxics-11-00073-f005:**
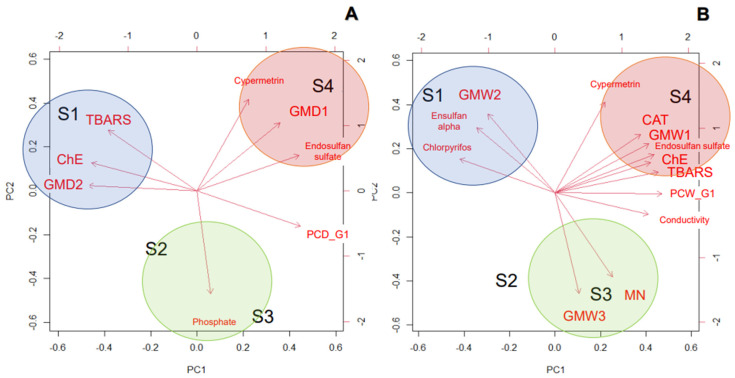
Biplot showing the differences among four sites in the Conlara River (Paso Grande—S1, Renca—S2, Concarán—S3, and Santa Rosa del Conlara—S4) due to physicochemical parameter, metal and pesticide contamination for the dry season (**A**) and the wet season (**B**). For the dry season, the variables are named as follows: GMD1 corresponds to arsenic and magnesium metals; GMD2 corresponds to cadmium, copper, barium, and lead metals, and; PCD_G1 corresponds to physicochemical variables correlated such as pH, conductivity, nitrate, nitrite, ammonia, total hardness, organic matter, dissolved oxygen, and DQO. For the wet season, the variables are named as follows: PCW_G1 corresponds to physicochemical variables correlated such as pH, nitrate, nitrite, ammonia, phosphate, total hardness, organic matter, and dissolved oxygen; GMW1 corresponds to arsenic and lead metals; GMW2: corresponds to chromium and zinc metals, and; GMW3 corresponds to iron and barium. The blue, red, and green circles represent the variables associated with Paso Grande (S1), Renca and Concarán (S2 and S3), and Santa Rosa del Conlara (S4), respectively.

**Table 1 toxics-11-00073-t001:** Environmental stressors (physicochemical, metals, and pesticides) measured in water samples at the four study sites (S1 to S4) in the Conlara River (San Luis, Argentina) in the dry and wet seasons.

		Site				Site				Permitted Level in Human Drinking Water (µg/L) ^a^	Permitted Level for Protection of Aquatic Life (µg/L) ^a^
	Analytic Measurement	S1	S2	S3	S4	S1	S2	S3	S4		
		Dry Season				Wet Season					
Physico Chemical	pH	8.77	8.42	9.01	7.86	8.92	7.66	7.81	7.85		
	Conductivity (µS/cm)	349	434	1680	1814	216	509	2142	1847		
	Nitrates (mg/L)	0.6	2.2	1.0	1.0	0.70	0.50	0.75	0.80	10,000	NL
	Nitrite (mg/L)	0.120	0.075	0.018	0.033	0.012	0.008	0.004	0.015	1000	60
	Amonia (mg/L)	0.04	0.13	0.60	0.60	0.21	0.36	0.16	0.19	50	1370
	Phosphate (mg/L)	0.028	0.021	0.023	0.028	0.012	0.185	0.020	0.062		
	Total Hardness (mg CaCO_3_/L)	176	180	468	484	61.45	136.8	529.1	471.9		
	DO (mg 0_2_/L)	11.23	10.16	10.16	8.32	8.50	6.22	9.26	8.93		
	COD (mg 0_2_/L)	3.58	3.87	2.72	2.58	1.40	1.40	1.26	1.55		
	BOD (mg 0_2_/L)	40.82	44.12	31.08	29.41	1.59	1.59	14.36	1.77		
Metals (µg/L)	As	6.27	3.48	5.32	21.09	7.13	6.37	11.04	23.34	50	50
	Ba	59.96	50.12	45.47	37.97	15.26	60.67	72.80	27.61	1000	NL
	Cd	0.441	0.031	0.001	0.031	ND	ND	ND	ND	5	0.2
	Cu	465.6	395.9	344.6	380.0	28.53	27.76	24.05	33.73	1000	2
	Cr	5.59	1.81	1.49	2.56	82.02	52.60	52.46	57.83	50	2
	Fe	618.32	95.71	113.44	142.10	194.03	501.31	407.98	269.13	NL	NL
	Mn	49.91	54.27	41.97	183.97	484.77	271.21	183.48	15.71	100	100
	Pb	50.01	0.42	0.90	ND	2.62	1.90	2.55	2.25	50	1
	Zn	52.29	45.96	58.36	24.67	100.4	38.59	32.47	51.93	5000	30
Pesticides (µg/L)	α-BHC	0.020	BLQs	0.025	0.020	0.019	0.019	0.020	0.025	0.131	0.01
	Chlorpyrifos	BLQs	BLQs	BLQs	BLQs	0.167	0.079	0.079	0.050	90	0.083
	Permetrin	0.177	BLQs	0.233	0.146	0.101	BLQs	BLQs	0.057	NL	0.004 ^b^
	Cypermethrin	0.271	0.108	0.322	0.183	0.132	0.108	0.116	0.107	NL	0.006
	Endosulfan α	BLQs	BLQs	0.107	BLQs	BLQs	BLQs	0.097	0.223	138	0.02
	Endosulfan β	0.222	BLQs	BLQs	BLQs	BLQs	BLQs	BLQs	BLQs		0.02
	Endosulfan sulfate	0.066	0.024	0.206	0.209	0.082	BLQs	BLQs	0.155		NL

BLQs: below limit of quantification. ND: not detected. NL: levels allowed by EPA or country regulations are not recorded. α-BHC: alpha-hexachlorocyclohexane. ^a^ Argentina normative based on EPA. ^b^ Canadian Water Quality. Guidelines for the Protection of Aquatic Life.

## Data Availability

Data is unavailable due to privacy or ethical restrictions.

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
