# Peer review of "Biomonitoring of Alterations in Fish That Inhabit Anthropic Aquatic Environments in a Basin from Semi-Arid Regions"

_toxics, 2023, doi:10.3390/toxics11010073_

Round 1

Reviewer 1 Report

TITLE: BIOMONITORING AND AQUATIC ENVIRONMENT FRAGMENTATION OF A BASIN FROM SEMI-ARID REGIONS AFFECTED BY ANTHROPOGENIC ACTIVITY

Dear Editor

In this study the authors have provided important information for Bio-monitoring in Aquatic Environment in Semi-Arid Regions Affected by manmade actions. The manuscript is organized and satisfactory but needs further improvement. However, it can be further improved; below I have provided few corrections. The manuscript requires major revisions.

1.      The abstract should be specific based on your findings. A short objective of study must be expressed in abstract.

2.      Please add recent literature in the introduction section as well as in the methodology section. Such as

Liu, M., Xu, Y., Rahman, Z., Khan, S., Idress, M., Ali, A., Ahmad, R., Khan, S.A., Khan, A. and Khan, M.Q., 2020. Contamination features, geo-accumulation, enrichments and human health risks of toxic heavy metal (loids) from fish consumption collected along Swat river, Pakistan. Environmental Technology & Innovation, 17, p.100554

Nawab J,  Khan S, Wang XP. (2018). Ecological and health risk assessment of potentially toxic elements in the major rivers of Pakistan: General population vs. Fishermen. (2018). Chemosphere. 202 ,154-164.

Nawab, J., Din, Z.U., Ahmad, R., Khan, S., Zafar, M.I., Faisal, S., Raziq, W., Khan, H., Rahman, Z.U., Ali, A. and Khan, M.Q., 2021. Occurrence, distribution, and pollution indices of potentially toxic elements within the bed sediments of the riverine system in Pakistan. Environmental Science and Pollution Research 28, 54986–55002 (2021). https://doi.org/10.1007/s11356-021-14783-9

3.      Please add a comparison table in your paper which will enhance the scientific quality of this paper.

4.      In figure 2 some error bars you have mentioned a and b, in some graph no captions while in some only a and b. Please clarify this in the figure legends.

5.      Your discussion needs further improvement and scientific logics. So improve it for international audience in a scientific manner.

6.      Your results need further clarification in terms of your study objectives and your paper title.

7.      If possible also calculate the health risk associated with the consumption of this fish for the local population

8.      A lot of studies have been conducted on this fish kindly add novelty of this study in comparison with other studies carried out on the same fish.

9.      Figure captions need to be describe your figure results (update figure captions)

10.  Please upgrade the quality of your figures and tables

11.  Please be consistent in this section (conclusion) according to your results and needs further improvement with respect to heavy metals accumulation in different regions.

Recommendations

I will review this paper again after incorporation of my comments

Author Response

  1. Response to reviewer: In total accordance with the reviewer, we restructured the manuscript by placing a shorter objective both in the abstract and in the main text.
  2. Response to reviewer: Following the reviewer's suggestions, references were added to both the introduction and discussion (LINES 37-43,91,334,415) references [7,21,54].

  3. Response to reviewer: We greatly appreciate the reviewer's suggestion to increase the quality of the work. However, we fail to understand what kind of comparison it refers to. In case of detailing that information, we will gladly add the table.

  4. Response to reviewer: Following the reviewer's suggestions, the legends and figure references were clarified. In particular, in figure 3 the units on the axes were added and the symbology changed. For figure 4, the appropriate symbology was added. In Figure 5, the letters of each graph were added to the legends. In addition, the symbology for statistical differences was modified and the letters for each figure were maintained.

  5. Response to reviewer: Following the reviewer's suggestions, some paragraphs were added in the discussion to improve and achieve the impact on the international audience, highlighting the focus of our work objective, which is to implement biomarkers and biomonitoring as complementary tools in aquatic environment quality assessments (LINES 331-336, 367-383, 413-429).

  6. Response to reviewer: We fully agree with the reviewer, in the modifications of the work we clarified our results based on the study objectives and even the title was modified considering this observation (LINES 331-336, 367-383, 413-429).

  7. Response to reviewer: We greatly appreciate the reviewer's suggestion to increase the quality of the work. We believe that the associated risk assessment is an excellent analysis to carry out and a point that we have proposed to address in the future, which we raise in the discussion. In any case, we consider that the data we have at the moment are insufficient to carry out such an analysis, it would be necessary to expand the measurements of environmental stressors in other ecotoxicological compartments such as sediments and biota (bioaccumulation). All these analyzes will be taken into account to carry them out in future biomonitoring.

  8. Response to reviewer: In total agreement with the reviewer, in the discussion we focused on the novel results that we found in this species of fish (LINES 331-336, 367-383).

  9. Response to reviewer: Following the reviewer's recommendations, we describe the titles of our figures as the results (LINES 659-680).

  10. Response to reviewer: Following the reviewer's recommendations, the quality of the figures and tables were improved.

  11. Response to reviewer: Taking this suggestion into consideration, in our discussion we refer to further improvements that should be made, not only in the region but also in other Neotropical regions (LINES 367-383; 413-429). However, we consider that our work does not only focus on metals, which is why we extend it to the rest of the environmental stressors evaluated by us. On the other hand, expanding the discussion to all environmental stressors would make this section very long.

Reviewer 2 Report

The field research by Pérez-Iglesias J.M and collaborators select Jenynsia multidentata as bio-monitoring specie and catch them from 4 different sample sites in Conlara River to investigate the adverse effects on different level endpoints. The novelty of the present work is that the authors integrate physio-chemical parameters,(in)organic pollutant concentrations and  bio-toxic endpoints to evaluate the eco-quality of local river. However, there are several problems that prevent my recommendation for publication in its current form. Major revisions are required before it can be considered for publication in Toxic.

Firstly, sampling sites are important variables in this work, representing different contamination states, but the current background information is insufficient. For example, if S4 is the most polluted point and there are agricultural pollution sources upstream of S4, can the pesticide usage be provided? Did the author gather some historical water quality monitoring information at the above points? In addition, the selection of monitored species is the highlight of this article. Please give more information about Jenynsia multidentata, such as appearance, life cycle and body length and weight at different stages.

Secondly, The Result should be an objective description of the experimental results. Possible reasons for the results should be included in Discussion. (For instance, line 323-324, 332-333). In addition, the current Discussion is not comprehensive enough. Only the antioxidant system and the results of PCA were discussed. The analysis of MNs and individual indicators is ignored. In other words, every result of the experiment should be analyzed in the discussion.

Format problems are the last concern. In manuscript, there are a lot of abbreviations but does not give the full name, which makes it difficult to read. Please check the full text to make sure the full name is given the first time the abbreviations are used. Besides, the style of figures and tables is rough and lacks concentration units.

The details are as below:

Abstract & Introduction

Line 23: It is recommended to give the specific name of the sampling sites. Moreover, please unify the names of the 4 sampling sites. There are at least 4 different types of names for the current manuscript.  "S1", "PG", "Paso Grande" and "Site1" are represent same place.

Materials and methods

Line 127-144:

More background information about different sites should be offered:

  1. The number of permanent residents in 4 towns
  1. Pesticides usage
  2. Tourists number in each research sites

Line 149: Give the full name of APHA, and list it as a reference.

Line 152: What's the meaning of S.M. 4500? I guess it a part of APHA. Please give reason why the authors chose APHA as the method? Please state which parameters are measured in the field and which are measured in the laboratory.

Line 161: Please include the standard in the references.

Line 172: Please include EPA 507 and 508 in the references.

Line 197: Although the authors state that the experiment complies with animal ethical requirements, the description of the method should clearly indicate whether the fish were anesthetized prior to dissection. In addition, "previous mentioned protocols" is a vague expression, please give a specific quote.

Line 201: In Evaluations of endpoints, how many fish were used in the tests for each endpoint? Please indicate in the corresponding position in the text.

Line211: In 2.4.2, the line 213 showed that livers were used as target organs to the follow biochemical tests, but in line 239, "the lipid peroxidation in whole fish was determined", did the determination of TBARS used the whole fish instead of livers?

Line 255: In Statistical analysis, the author should state which form of error were used, mean ± SD or mean ± SEM? If the former is used, do not change it. If the latter is used, change it to mean ± SD, as the latter options are more visually representative of the statistics used and will allow the reader a visual representation of the spread of the data.

Results

Line 285: Although there is no significant difference at individual level, please consider adding corresponding figure. Because this is the only result that represents the individual level response in this work.

Line 308: please add the results of MNs in dry season.

Line 313: In this section, the sudden appearance of some terms is confusing:

Line 317: The authors use "the Water Quality Index (WQI)" to integrate several Physical and chemical parameters, but there is no description about how to calculate WQI in Methods.

Line 327,333,340,343: What are the meanings of GM1/GM2/GM3? And why the GM1 represent different parameters in different seasons?

Discussion

Line 410-414:  As the author said, the survival pressure of fish is greater in heavily polluted areas, a question is why did the author not make statistics on the amount of catch in different sampling sites when designing the experiment? In addition, what is the swimming range of the fish and whether there is cross-sites swimming activity?

Line 414-417: The authors emphasize that the fish selected are suitable indicator species, but comparisons with other fish are lacking. Please consider adding literature for cross-sectional comparisons.

Figures and Table:

Fig1-A: The circle simply marks the research scope. Please consider further refining the research boundaries.

Fig1-B: Please mark the river direction and add the scale.

Table 1 integrates the information of three types of indicators, which can be divided into three tables. Another suggestion is to give the measurement units of each indicator in the table.

Figure 2-3: If use Tukey test for one way ANOVA, the author should add letters in every groups.

The Tukey test compares every mean with every other mean. Absolute different letters between 2 groups indicates the significant differences. If there has repeat letters between 2 groups, there is no significant difference.

Add units on the y-axis for different test endpoints.

In Figure2E and Figure2F, the number format on the y axis should be consistent.

Figure 4: Check all the abbreviations carefully and explain them in caption. For ease of reading, Figure 4 should also first depict the "wet season" and then the "dry season", the same as shown in Figure 2-3.

Language

An additional inspection for English language is required prior to resubmission. Some expressions are colloquial. For instance, in line 197, "slaughter" ("dissection" may be better). Line 283: "all …biomarkers …very useful".

Author Response

We appreciate the reviewer's suggestions that undoubtedly improved our work, below we raise our responses:

Abstract & Introduction

Line 23: It is recommended to give the specific name of the sampling sites. Moreover, please unify the names of the 4 sampling sites. There are at least 4 different types of names for the current manuscript.  "S1", "PG", "Paso Grande" and "Site1" are represent same place.

Response to reviewer: We appreciate the suggestion. The names of the sites were standardized throughout the manuscript with the same symbology.

Materials and methods

Line 127-144: More background information about different sites should be offered:

The number of permanent residents in 4 towns, Pesticide’s usage, Tourists number in each research sites

Response to reviewer: In full agreement with the reviewer, we added important information about the cities (LINES 106-119).

Line 149: Give the full name of APHA, and list it as a reference.

Response to reviewer: This suggestion was taken into account, the reference was prepared and clarified in the text (LINE 125)

Line 152: What's the meaning of S.M. 4500? I guess it a part of APHA. Please give reason why the authors chose APHA as the method? Please state which parameters are measured in the field and which are measured in the laboratory.

Response to reviewer: As requested by the reviewer, the detail on the measurement of parameters in the laboratory was added (LINES 127-128). On the other hand, we note that the acronym S.M. refers to “Standart Methods” and the number with its corresponding letter refers to the particular methodology proposed by APHA to estimate that parameter. Finally, we use the APHA guidelines because they are the worldwide standardized measures to estimate the physicochemical parameters, which also makes laboratory quality controls.

Line 161: Please include the standard in the references.

Response to reviewer: We appreciate the reviewer's comment, the references were included.

Line 172: Please include EPA 507 and 508 in the references.

Response to reviewer: We appreciate the reviewer's comment, the references were included (LINES 139-141).

Line 197: Although the authors state that the experiment complies with animal ethical requirements, the description of the method should clearly indicate whether the fish were anesthetized prior to dissection. In addition, "previous mentioned protocols" is a vague expression, please give a specific quote.

Response to reviewer: We accept the reviewer's suggestion and make the corresponding changes (LINES 166-169).

Line 201: In Evaluations of endpoints, how many fish were used in the tests for each endpoint? Please indicate in the corresponding position in the text.

Response to reviewer: Taking this suggestion into account, the number of fish used by sites was added in the section “2.3 Test organisms and specimen collection” (LINE 149). It should be noted that all the mentioned endpoints were evaluated in the same individual.

Line 211: In 2.4.2, the line 213 showed that livers were used as target organs to the follow biochemical tests, but in line 239, "the lipid peroxidation in whole fish was determined", did the determination of TBARS used the whole fish instead of livers?

Response to reviewer: In full agreement with the reviewer, this point was clarified (LINES 187-188).

Line 255: In Statistical analysis, the author should state which form of error were used, mean ± SD or mean ± SEM? If the former is used, do not change it. If the latter is used, change it to mean ± SD, as the latter options are more visually representative of the statistics used and will allow the reader a visual representation of the spread of the data.

Response to reviewer: In full agreement with the reviewer, this point was clarified (LINES 662-672).

Results

Line 285: Although there is no significant difference at individual level, please consider adding corresponding figure. Because this is the only result that represents the individual level response in this work.

Response to reviewer: Taking this suggestion into account, Figure 2 was added, which represents the graphs of the individual level.

Line 308: please add the results of MNs in dry season.

Response to reviewer: Considering this point, we add the MNs frequency in dry season to Figure 4

Line 313: In this section, the sudden appearance of some terms is confusing:

Response to reviewer: Taking this suggestion into account, below are the responses to these confusing terms.

Line 317: The authors use "the Water Quality Index (WQI)" to integrate several Physical and chemical parameters, but there is no description about how to calculate WQI in Methods.

Response to reviewer:  In full agreement, this analysis was withdrawn because it was not an objective of our work, it was only carried out in multivariate analysis.

Line 327,333,340,343: What are the meanings of GM1/GM2/GM3? And why the GM1 represent different parameters in different seasons?

Response to reviewer: Considering this point, we add the clarification on these acronyms (LINES 299-322). On the other hand, if the reviewer considers it necessary, we can add the correlation matrices endorsing this resolution taken by the authors to reduce the variables.

Discussion

Line 410-414:  As the author said, the survival pressure of fish is greater in heavily polluted areas, a question is why did the author not make statistics on the amount of catch in different sampling sites when designing the experiment? In addition, what is the swimming range of the fish and whether there is cross-sites swimming activity?

Response to reviewer: We thank you for the suggestion. Added to the discussion is the proposal to make estimates of abundance and richness of fish species to assess the direct impact at the population level (LINES 410-413). Unfortunately, we do not have information on the swimming range of the species, so we do not add that part to the discussion.

Line 414-417: The authors emphasize that the fish selected are suitable indicator species, but comparisons with other fish are lacking. Please consider adding literature for cross-sectional comparisons.

Response to reviewer: In total agreement with the reviewer's observation, the discussion in comparison with other bioindicator species proposed for the Neotropical region (LINES 367-383) was added to the manuscript. However, we highlight that the selection of the species, in addition to the characteristics mentioned in the section "2.3 Test organisms and specimen recollection", is due to the fact that it is the only species present in all the sampling sites, this was corroborated by previous studies in the area (LINES 159-161). Finally, we highlight that there are many studies carried out on bioindicator species proposed for the Neotropical region; however, these focus on laboratory bioassays with Cnesterodon decemmaculatus and not on biomonitoring as proposed in our work. Finally, we mention that we have not made comparisons with benthic fish species since the route of exposure to a greater degree is given by the sediment or with species used for human consumption since fishing occurs in specific regions of the basin, only in dams.

Figures and Table:

Fig1-A: The circle simply marks the research scope. Please consider further refining the research boundaries.

Response to reviewer: Based on the reviewer's comment, the quality of Figure 1 was improved and the sampling area was refined.

Fig1-B: Please mark the river direction and add the scale.

Response to reviewer: Based on the reviewer's comment, the river direction and the scale in Figure 1 was marked.

Table 1 integrates the information of three types of indicators, which can be divided into three tables. Another suggestion is to give the measurement units of each indicator in the table.

Response to reviewer: Taking into account the reviewer's suggestion, we give the units of measurement of each indicator in the table.

Figure 2-3: If use Tukey test for one way ANOVA, the author should add letters in every groups.

The Tukey test compares every mean with every other mean. Absolute different letters between 2 groups indicates the significant differences. If there has repeat letters between 2 groups, there is no significant difference.

Response to reviewer: based on the reviewer's comment, we clarify in the legends of the figures the symbology corresponding to the significant differences of the Tukey test between the groups (LINES 667-668, 671).

Add units on the y-axis for different test endpoints.

Response to reviewer: Taking into account the reviewer's suggestion, we give the units of the y-axis for test endpoints.

In Figure2E and Figure2F, the number format on the y axis should be consistent.

Response to reviewer: In full agreement with the reviewer's suggestion, we give the units for Figure 2E and 2F.

Figure 4: Check all the abbreviations carefully and explain them in caption. For ease of reading, Figure 4 should also first depict the "wet season" and then the "dry season", the same as shown in Figure 2-3.

Response to reviewer: In total agreement with the reviewer's suggestion, we check all the abbreviations carefully and explain them in caption for ease of reading (LINES 673-684).

Language

Response to reviewer: We accepted both the editor's and the reviewer's suggestion and the paper was sent for careful language review by a native speaker and a language correction service provided by the University.

An additional inspection for English language is required prior to resubmission. Some expressions are colloquial. For instance, in line 197, "slaughter" ("dissection" may be better). Line 283: "all …biomarkers …very useful".

Response to reviewer: In total agreement with the reviewer, we reviewed the entire manuscript and these sentences inconsistent with the language were modified.

Round 2

Reviewer 1 Report

Revise the graphs it has some problem specially figure (1 and 5)

Author Response

Reviewer #1: Revise the graphs it has some problem specially figure (1 and 5).

Response: In full agreement with the reviewer's request, Figures 1 and 5 were reviewed. In addition, modifications were made to both for a better understanding of the graphic representation and more detailed legends were added to help understand what is represented (LINES 126-130 and 303-313).

Reviewer 2 Report

The author replied all questions, I think that this manuscript is ok in this present form. Therefore, I have no other question.

Author Response

No comments or suggestions were made by the reviewer.
